# The Effects of Learning and Eating Behaviours among Medical Students during the COVID-19 Pandemic

Sayeeda Rahman [1,*], Rita Kirton [1], Brenda Roach [1], Maria Teresa Villagomez Montero [1], Alexey Podcheko [1], Nader Nouraee [1], Sadia Ahsan [1], Joshua Noel Nisar [1] and Ahbab Mohammad Fazle Rabbi [2]

[1]  School of Medicine, American University of Integrated Sciences School of Medicine, Bridgetown BB 11114, Barbados
[2]  Department of Population Sciences, University of Dhaka, Dhaka 1000, Bangladesh
*   Correspondence: srahman@auis.edu

**Abstract: Background:** The effect of the COVID-19 pandemic has transformed medical education and is likely to have long-lasting effects on student learning, mental well-being, and eating behaviour. This study aimed to examine the learning behaviours of medical students at the American University of Integrative Sciences (AUIS), Barbados, during the COVID-19 pandemic. **Methods:** A cross-sectional web-based on-line survey was administered to medical students at AUIS from July to November 2021. The data collecting instrument recorded students' demographic and learning behaviour information and eating disorders (SCOFF questionnaire). **Results:** The overall response rate was 55% ($n$ = 120). In relation to learning behaviour, students agreed with the following statements: 'deterioration in work performance and studying' (48.4%), 'remember subject's contents appropriately' (40.4%), 'concentration on the studies' (40.3%), 'difficulty in performing two tasks simultaneously' (38.7%), 'difficulty in performing mental calculations' (33.9%), 'difficulty in recalling recent information' (32.3%), and 'difficulty in recalling old information' (38.7%). Among the eight dimensions of learning behaviours, deterioration in work performance or studying and difficulties in recalling recent information were found to be significantly associated with the gender of the students. For the SCOFF questionnaire, approximately 24.2% screened positive for eating disorders. Screening with the SCOFF test demonstrated that females, older (>25 years), overweight + obese, Clinical Sciences + PreMed, and non-USA-based students were at more risk of eating disorders. **Conclusions:** The results indicate that during the COVID-19 pandemic, AUIS students have developed learning difficulties and are likely to have eating disorders. University policymakers should take appropriate measures to support a healthy learning environment and improve students' mental well-being and eating behaviours.

**Keywords:** learning behaviours; eating disorders; SCOFF questionnaire; medical students; COVID-19 pandemic; Barbados

## 1. Introduction

The COVID-19 pandemic caused many countries worldwide to implement lockdown/quarantine policies, with most universities announcing a teaching pause to protect staff and student health and safety by minimizing the spread of this highly contagious disease [1–3]. Coronavirus disease 2019 (COVID-19) is a very transmittable and infectious disease caused by a new strain of severe acute respiratory syndrome Coronavirus-2 (SARS-CoV-2) [4–6]. Coronaviruses are RNA viruses that belong to the Coronaviridae family. They contain large RNA virus genomes, are highly replicative due to conserved genomic organization, show several enzymatic activities, and have extensive ribosomal frameshifting due to numerous non-structural gene expressions [4]. Some coronaviruses that infect animals can undergo mutation and adaptation, which drives the coevolution of coronaviruses to become new human coronaviruses (HCoVs) [5]. The first human coronaviruses were

identified in the mid-1960s. SARS-CoV and MERS-CoV are responsible for Severe acute respiratory syndrome and Middle East respiratory syndrome, respectively. Later there was SARS-CoV-2 in 2019 or COVID-19. These HCoV infections are zoonotic [4,5]. Based on the genetic analysis of the virus, it has been identified as a closely related coronavirus to those found in bats, which was transmitted to human subjects through an intermediate host, such as a pangolin, at a wildlife market in Wuhan, China [5]. However, there is still ongoing research and investigation on the origin of COVID-19 [7].

The usual signs and symptoms of COVID-19 are similar to the common cold, accompanied by mild to moderate upper respiratory tract infection (URTI). The disease is primarily transmitted from person to person through respiratory droplets when an infected person talks, coughs, or sneezes [8]. Implementation of social distancing, suspension of face-to-face teaching, and travel limitations caused a disruption within the education sector. Infection control policies mandated the closure of schools and universities [9], which has greatly impacted the mental and physical health of the students [10]. As reported recently, specifically, the pandemic has affected problematic eating behaviours amongst young adults and students through multiple pathways [11], the emotional impact of the restrictions being the foremost, loneliness and social isolation, boredom, and decreased physical and social activities [12]. Thus, pandemic-imposed measures may result in major psychological negative consequences such as post-traumatic stress symptoms, negative effects, and anger, posing a threat to health and affecting life in general and psychological well-being in particular [11].

Furthermore, the lack of physical presence of peers can lead to negative learning experiences for some students, compounded by increased anxiety and depression with constant news updates and the dynamic circumstances of the pandemic [12]. Medical and allied health professional students are also more susceptible to stress due to the sudden shift to on-line learning and lack of social and peer interaction [9,13–15]. Such uncertainties and stressful situations create barriers to learning, as well as loneliness due to social distancing contributes to a further increase in stress, anxiety, and problematic eating behaviours. While some students found remote learning beneficial as it created flexibility and comfort [16], others found it quite challenging [17]. These challenges included decreased supervision by educators, difficulty learning within the home setting, and the notion of self-learning [18]. Identifying how these obstacles have impacted students learning behaviour is both valuable and critical. Studies have demonstrated that the overarching use of technology provided emotional support for students in the form of direct professor-student interactions, and the stimulation of real-time feedback allowed education to be focused on the student [19,20]. Educational models that emphasize such factors will likely improve on-line learning behaviour.

The COVID-19 pandemic and related limitations have negatively impacted the food habits of the majority of the world's population [21]. Due to a decrease in food availability, food acquisition, and food quality, several South American, African, and Asian nations have experienced food insecurity [22–24]. Changes in dietary habits have occurred in North America and Europe, yet food insecurity has not increased proportionally [25]. The confinement has also caused an increase in psychological discomfort, leading to increased emotional eating, weight gain, a drop in physical activity, and an overall rise in sedentary living [26,27].

According to Coakley et al. [28], University students with symptoms of moderate to severe anxiety reported higher degrees of maladaptive appetitive traits: hunger, food, and satiety responsiveness, and emotional over-eating, with a decrease in enjoyment of food [9]. Although women were more likely to consume fresh foods, many experienced more psychological distress with higher levels of emotional eating that led to weight gain [25,29]. Among student populations, there has not been much change in dietary and lifestyle habits, but different student populations have experienced food insecurity based on their socio-demographic location and the availability of food in their country [27,30,31]. Depending on the change in living situations, some students increased their consumption of healthy

foods and physical activity while others continued to order take-out and increased their overall food consumption. Most student populations had a reduction in the consumption of fresh foods with an overall increase in alcohol consumption and smoking [25].

The American University of Integrative Sciences (AUIS), Barbados, has a mixed population of students from many parts of the world. Although most students are from the US, many come from India, Nigeria, Canada, the Caribbean Islands, and Europe. Depending on their country of origin, the AUIS medical students adopt a mixture of both healthy and unhealthy dietary lifestyles. While most AUIS students reported daily eating habits involving breakfast, lunch, and dinner, some described irregular dietary arrangements, such as skipping all meals except dinner. Students with irregular eating habits stated mostly taking coffee, energy drinks, sugary drinks, and unhealthy snacks such as chips and chocolate. Many students have also reported weight gain due to their dietary lifestyle. Therefore, it is evident that during the COVID-19 pandemic, students have experienced various changes in their dietary habits, with some improving and others decreasing in their nutrition intake [32].

This study aims to examine the learning and eating behaviour among medical students of the American University of Integrative Sciences (AUIS) in Barbados during the COVID-19 pandemic.

## 2. Materials and Methods

### 2.1. Study Design, Sampling, and Data Collection

*Study type:* A Cross-sectional study was conducted at AUIS from July until November 2021.

*Study Participants and procedures:* All AUIS PreMed and medical students were invited to participate in this study. A validated on-line learning questionnaire was published using the on-line Google Form link, with those students willing to participate completing an informed consent form. The consent form explained the purpose of the study to the participants. Participation was voluntary, with participants given the option to withdraw at any time during data collection. The questionnaire was anonymous, with no identifying information being collected. The questionnaire was piloted with five students (2 PreMed and 3 Med), and they were not included in the main survey.

*Inclusion criteria:* All AUIS PreMed and medical students.
*Exclusion criteria:* Students who declined to participate in the survey.
*Study instrument:* These data were collected using the following study tools:

1. Students' demographic information
2. Learning behaviour information: A validated on-line learning questionnaire on Learning Behaviour information was utilized [3]. The questionnaire assessed the impact of quarantine on students' learning behaviours with a Five Point Likert Scale (strongly disagree to strongly agree) used to collect the information. The survey questionnaire consisted of 8 items.
3. Screening for eating disorders (SCOFF): The SCOFF questionnaire is a five-item screening questionnaire for eating disorders developed by Morgan et al. [33]. The questionnaire is designed to examine whether an eating disorder is present rather than to make a diagnosis and investigate the key aspects of eating disorders, i.e., vomiting, concerns about losing control over how much one eats, weight loss, feeling fat, and whether food dominates life. These questions can be answered by 'yes' or 'no.' Scores of 2 or greater indicate that the participant could have a high possibility of having anorexia nervosa or bulimia nervosa.

### 2.2. Ethical Approval

Ethical approval was obtained from the research committee of the American University of Integrative Sciences (AUIS-RC/2021/R#02-SR). The study was conducted by the guidelines of the 1975 Declaration of Helsinki.

*2.3. Statistical Analysis*

All statistical analysis was performed using (SPSS) software version 22.0 for Microsoft windows. The numbers and percentages were calculated. A *p*-value < 0.05 was considered to be significant. Reporting descriptive statistics about student knowledge/perceptions of COVID-19 and vaccine acceptance [means and percentage]. Informing descriptive statistics for items measuring students' perceived stress and learning behaviour of students [means and percentage].

**3. Results**

The response rate was 52% (*n* = 62). The majority of the respondents were females (59.7%) (Table 1). Most of the respondents (56.5%) were residing in the USA during the survey. The mean age of the participants was 31.84 years (SD = 8.68). The percentage of participants within the normal range of BMI was 50%. The percentage of those who were underweight was 4.8%, and 25.8% of the participants were classified as overweight by the BMI classification of the WHO. More than three fourth of the respondents (87%) received the COVID-19 vaccine, and only 13% tested COVID-19 positive during the survey period.

**Table 1.** Medical students' Socio-demographic Characteristics (*n* = 62).

| Socio-Demographic Characteristics | Responses (%) |
|---|---|
| *Gender* | |
| Male | 25 (40.3%) |
| Female | 37 (59.7%) |
| *Age (Year ± SD)* | |
| Male | 32.64 ± 8.72 |
| Female | 31.3 ± 8.73 |
| Total | 31.84 ± 8.68 |
| *Body Mass Index* | |
| Underweight | 3 (4.8%) |
| Normal | 31 (50%) |
| Overweight | 16 (25.8%) |
| Obesity | 12 (19.4%) |
| *Level of study* | |
| Clinical Sciences Yr 1 | 11 (17.7%) |
| Clinical Sciences Yr 2 | 17 (27.4%) |
| PreMed 1 | 2 (3.2%) |
| PreMed 2 | 3 (4.8%) |
| MD 1 | 3 (4.8%) |
| MD 2 | 4 (6.5%) |
| MD 3 | 3 (4.8%) |
| MD 4 | 4 (6.5%) |
| MD5 | 15 (24.2%) |
| *Location of respondent* | |
| USA | 35 (56.5%) |
| Canada | 14 (22.6%) |
| India | 4 (6.5%) |
| Barbados | 4 (6.5%) |
| Others (Japan, Jamaica, Trinidad, etc.) | 5 (8.1%) |

Regarding learning behaviour, students agreed with the following statements: 'deterioration in work performance and studying' (48.4%), 'remember subject's contents appropriately' (40.4%), 'concentration on the studies' (40.3%), 'difficulty in performing two tasks simultaneously' (38.7%), 'difficulty in performing mental calculations' (33.9%), 'diffi-

culty in recalling recent information' (32.3%), and 'difficulty in recalling old information' (38.7%) (Table 2).

The associations between sex and the pandemic's effect on students' learning behaviours at AUIS are summarized in Table 2. Responses for the outcomes related to learning behaviours are summarized into three categories: disagree (containing strongly disagree and disagree), neutral, and agree (containing both agreed and strongly agreed respondents). Among the eight dimensions of learning behaviours, only noticing a deterioration in work performance or studying and difficulties in recalling recent information were found to be significantly associated with the sex of the students. As per the structure of the sample, female frequencies were higher for all three categories of learning behaviours; however, exceptionally higher percentages were observed for female students who significantly noticed a deterioration in work performance or studying (37.1%).

The relationship between students' study levels and learning behaviours at AUIS was also examined, which is summarized in Table 3. None of the dimensions of learning behaviours showed a significant association with the student's study levels.

Regarding the SCOFF questionnaire, 24.2% of the participants reported two or more yes-responses (SCOFF test positive). Among 62 participants, 24 (38.7%) answered 'yes' to the SCOFF question 'Do you worry you have lost control over how much you eat?', while only 3 (4.8%) participants answered 'yes' to the question 'Do you ever make yourself sick (vomit) because you feel uncomfortably full?' (Table 4). Screening with the SCOFF test demonstrated that females, older (>25 years), overweight + obese, Clinical Sciences + PreMed, and non-USA-based students were at more risk of eating disorders (Table 5).

We also checked the association between eating disorders and all eight dimensions of learning behaviour information. There was no significant relationship between any learning behaviour variables and eating disorders. All the indicators of learning behaviours fall into three categories: disagree (which also includes strongly disagreed respondents), neutral, and agree (which includes strongly agreed respondents). More than 29% of the respondents who were SCOFF tested positive had an impact on their learning behaviour during the COVID-19 pandemic (Figure 1). On the contrary, only 21.6% of SCOFF test-negative students had experienced learning difficulties (Figure 2).

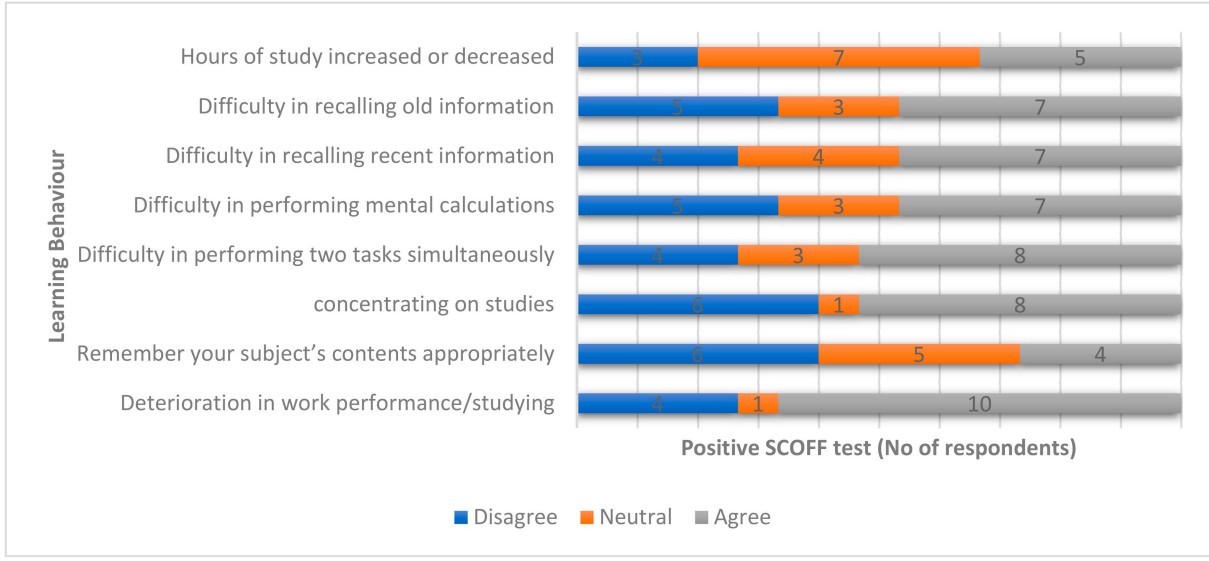

**Figure 1.** Association between a positive SCOFF test and learning behaviour(No statistically significant relationship was observed).

**Table 2.** Effect of COVID-19 pandemic on learning behaviours among medical students of AUIS ($n = 62$) vs. gender of respondents.

| | | Strongly Disagree + Disagree | | Neutral | | Agree + Strongly Agree | | *p*-Value |
|---|---|---|---|---|---|---|---|---|
| | | Male | Female | Male | Female | Male | Female | |
| 1. | Have you noticed deterioration in your work performance/studying? | 11 (17.7%) | 10 (16.1%) | 7 (11.3%) | 4 (6.5%) | 7 (11.3%) | 23 (37.1%) | 0.025 ** |
| 2. | Do you remember your subject's contents appropriately? | 5 (8.1%) | 9 (14.5%) | 8 (12.9%) | 15 (24.2%) | 12 (19.4%) | 13 (21%) | 0.598 |
| 3. | Are you appropriately concentrating on your studies? | 7 (11.3%) | 14 (22.6%) | 5 (8.1%) | 11 (17.7%) | 13 (21%) | 12 (19.4%) | 0.303 |
| 4. | Are you having difficulty in performing two tasks simultaneously | 9 (14.5%) | 15 (24.2%) | 7 (11.3%) | 7 (11.3%) | 9 (14.5%) | 15 (24.2%) | 0.703 |
| 5. | Are you having difficulty in performing mental calculations? | 14 (22.6%) | 15 (24.2%) | 4 (6.5%) | 8 (12.9%) | 7 (11.3%) | 14 (22.6%) | 0.489 |
| 6. | Are you having difficulty in recalling recent information? | 14 (22.6%) | 10 (16.1%) | 5 (8.1%) | 13 (21%) | 6 (9.7%) | 14 (22.6%) | 0.071 * |
| 7. | Are you having difficulty in recalling old information? | 13 (21%) | 15 (24.2%) | 2 (3.2%) | 8 (12.9%) | 10 (16.1%) | 14 (22.6%) | 0.338 |
| 8. | Are the hours of study increased or decreased? | 5 (8.1%) | 6 (9.7%) | 14 (22.6%) | 14 (22.6%) | 6 (9.7%) | 17 (27.4%) | 0.207 |

Note: Significance: *** $p < 0.01$, ** $p < 0.05$, * $p < 0.1$.

**Table 3.** Effect of COVID-19 pandemic on learning behaviours among medical students of AUIS ($n = 62$) vs. study level of respondents (medical/pre-med).

| | | Strongly Disagree + Disagree | | Neutral | | Agree + Strongly Agree | | *p*-Value |
|---|---|---|---|---|---|---|---|---|
| | | Pre-Med | Medical | Pre-Med | Medical | Pre-Med | Medical | |
| 1. | Have you noticed deterioration in your work performance/studying? | 11 (17.7%) | 10 (16.1%) | 7 (11.3%) | 4 (6.5%) | 15 (24.2%) | 15 (24.2%) | 0.826 |
| 2. | Do you remember your subject's contents appropriately? | 8 (12.9%) | 6 (9.7%) | 10 (16.1%) | 13 (21%) | 15 (24.2%) | 10 (16.1%) | 0.490 |
| 3. | Are you appropriately concentrating on your studies? | 11 (17.7%) | 10 (16.1%) | 7 (11.3%) | 9 (14.5%) | 15 (24.2%) | 10 (16.1%) | 0.593 |
| 4. | Are you having difficulty in performing two tasks simultaneously | 9 (14.5%) | 15 (24.2%) | 10 (16.1%) | 4 (6.5%) | 14 (22.6%) | 10 (16.1%) | 0.105 |
| 5. | Are you having difficulty in performing mental calculations? | 16 (25.8%) | 13 (21%) | 5 (8.1%) | 7 (11.3%) | 12 (19.4%) | 9 (14.5%) | 0.664 |
| 6. | Are you having difficulty in recalling recent information? | 12 (19.4%) | 12 (19.4%) | 9 (14.5%) | 9 (14.5%) | 12 (19.4%) | 8 (12.9%) | 0.762 |
| 7. | Are you having difficulty in recalling old information? | 15 (24.2%) | 13 (21%) | 5 (8.1%) | 5 (8.1%) | 13 (21%) | 11 (17.7%) | 0.974 |
| 8. | Are the hours of study increased or decreased? | 7 (11.3%) | 4 (6.5%) | 16 (25.8%) | 12 (19.4%) | 10 (16.1%) | 13 (21%) | 0.466 |

Note: Significance: *** $p < 0.01$, ** $p < 0.05$, * $p < 0.1$.

**Table 4.** SCOFF questionnaire—Analysis of the findings.

| Item No. of The SCOFF | Number of 'Yes' Responses ($n$ = 62) | Percentage |
|---|---|---|
| 1. Do you ever make yourself sick (vomit) because you feel uncomfortably full? | 3 | 4.8% |
| 2. Do you worry you have lost control over how much you eat? | 24 | 38.7% |
| 3. Have you recently lost more than one stone (approx. 6 Kg) in three months period? | 6 | 9.7% |
| 4. Do you believe yourself to be fat when others say you are too thin? | 11 | 17.7% |
| 5. Would you say that food dominates your life? | 9 | 14.5% |
| **SCOFF score** | | |
| 0-yes-response | 31 | 50.0% |
| 1-yes-response | 16 | 25.8% |
| 2-yes-responses | 10 | 16.1% |
| 3-yes-responses | 4 | 6.5% |
| 4-yes-responses | 0 | 0% |
| 5-yes-responses | 1 | 1.6% |

**Table 5.** SCOFF results in relation to gender, age, BMI, level of study, and location.

| Variables | SCOFF Test Positive-Scored More than 2 and above | SCOFF Test Negative |
|---|---|---|
| | **Responses (%)** | |
| *Gender* | | |
| Male | 4 (6.5%) | 21 (33.9%) |
| Female | 11 (17.7%) | 26 (41.9%) |
| *Age (Year ± SD)* | | |
| ≤25 years | 5 (8.1%) | 10 (16.1%) |
| >25 years | 10 (16.1%) | 37 (59.7%) |
| *Body Mass Index (BMI)* | | |
| Underweight | 0 | 3 (5.2%) |
| Normal | 5 (8.6%) | 24 (41.4%) |
| Overweight | 5 (8.6%) | 11 (19%) |
| Obese | 4 (6.9%) | 6 (10.3%) |
| *Level of study* | | |
| Clinical Sciences + PreMed | 9 (14.5%) | 24 (38.7%) |
| MD1–MD5 | 6 (9.7%) | 23 (37.1%) |
| *Location of respondent* | | |
| USA-based | 5 (8.1%) | 30 (48.4%) |
| Non-USA based | 10 (16.1%) | 17 (16.1%) |

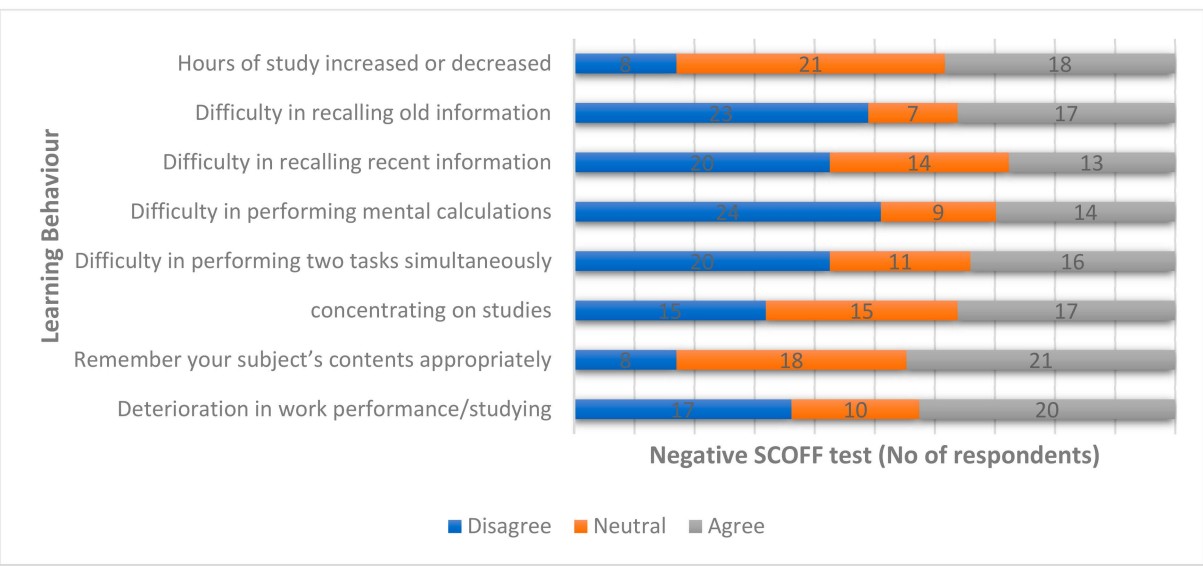

**Figure 2.** Association between a negative SCOFF test and learning behaviour (No statistically significant relationship was observed).

## 4. Discussion

This study attempted to identify the impact of COVID-19 on undergraduate medical students at AUIS. The findings demonstrated that AUIS students experienced problems in their learning behaviour and eating disorders during the COVID-19 pandemic.

Experiencing quarantine conditions has strained many individuals, including medical students [1–3,14]. Medical students' academic and professional performance has dropped due to the epidemic [3]. In our study, one-third to less than half of AUIS students reported a decline in their ability to study as well as difficulty in memorizing topic material, focusing on their studies, performing mental calculations, and recalling old and recent information. Students struggled to adapt to home-based instruction [1,2,9]. Prolonged screen exposure during on-line education may create brain changes that affect the focus, language intelligence, and processing speed, which may substantially impact learning processes [34]. Consequently, on-line learning during the COVID-19 pandemic, which mostly relies on computer-based platforms and necessitates more screen time, may have comparable effects on students' learning patterns [35].

During the COVID-19 pandemic, the learning behaviour of medical students has been influenced by elements including remote learning, reduced social engagement, and heightened stress [3,35]. Numerous students have had to acclimate to on-line classes, which can be tough owing to technological issues, lack of motivation, and home distractions. Moreover, the lack of face-to-face connection with peers and instructors has made it more difficult for certain students to remain motivated and focused [36]. In addition, elevated levels of stress brought on by the pandemic and its effects on their life can have a negative impact on their learning behaviour. During the COVID-19 epidemic, Curelaru et al. [35] found learning process issues among undergraduate and postgraduate university students in Romania, which included misunderstandings, a lack of feedback, increased academic obligations, a lack of challenge, and disengagement. In addition, Almendingen et al. [36] found a general sense of decreased motivation and effort across students' perceptions of on-line learning during the COVID-19 epidemic. A survey conducted in Pakistan demonstrated that 77% of medical students viewed on-line learning negatively in light of the present epidemic [37]. Similarly, more than one quarter (26.8%) of the medical students in Jordan expressed overall dissatisfaction with on-line learning [38].

However, some students have adapted well to the new learning environment and discovered strategies to remain interested and involved. The COVID-19 pandemic has also had a substantial impact on the clinical training and learning environment of students,

making it more challenging for them to attain practical experience and practice clinical skills [39,40]. The lack of in-person clinical rotations has also restricted their exposure to patients and the medical profession. A study in Libya revealed that 54.1% of students believed that on-line learning could facilitate interactive learning; however, only 21% felt that clinical learning could be successfully achieved through the use of technological methods [41]. In addition, medical students have had to adapt to changes in curriculum and assessment methods since some schools have used on-line tests and remote proctoring [42]. The additional stress and anxiety induced by the epidemic may further impact their learning behaviour [19]. However, medical students have demonstrated resilience and adaptability, with many finding inventive ways to complement their education, including virtual patient simulations and telemedicine encounters [15,43].

In addition, the COVID-19 pandemic has had substantial effects on students, including increased stress and changes in lifestyle and dietary habits [44,45]. Disruptions in classes/clinical rotations, postponed exams, and remote teaching during the COVID-19 pandemic contributed to higher stress and anxiety levels among medical students. As medical students witnessed the catastrophic impacts of the pandemic in healthcare settings, the prevalence of mental illness among medical students was found to be relatively higher than in the general population [46]. These changes and challenges may lead to the onset or worsening of eating disorders such as anorexia, bulimia, and binge eating disorder [47]. Furthermore, the epidemic has impeded access to mental health care and support. Students who are struggling with stress or disordered eating should emphasize self-care, seek support, and seek assistance [48]. In this survey, 45.2% of students were determined to be fat or overweight [49]. Previous research has demonstrated a comparable prevalence of obesity during pandemics. Approximately one-quarter of the AUIS student population had a positive SCOFF test, i.e., those students are at more risk of developing eating disorders [50]. It is very concerning that more than 38% of participants responded affirmatively to the second SCOFF question (Do you worry that you have lost control over how much you eat?). During the COVID-19 pandemic lockdown, Flaudias et al. [11] also found that 38.3% students were SCOFF positive, i.e., they were more likely to engage in hazardous eating behaviours during COVID-19 pandemic lockdown. Female students in our study were more susceptible to eating disorders which were supported by studies conducted during the COVID-19 pandemic by Tavolacci et al. [51] and Mahar et al. [52].

Stress and anxiety produced by the COVID-19 pandemic have a negative effect on mental health, and it appears to contribute to eating patterns [53,54]. Chan & Chiu [55] found that participants with suspected eating disorders reported significantly greater levels of depression and anxiety symptoms, as well as lower levels of three categories of psychological well-being (environmental mastery, purpose in life, and self-acceptance). These findings indicated that eating disorders require increased clinical attention during the COVID-19 pandemic.

*Limitations*

This study had a number of limitations. One of the important limitations of the present study was the small sample size and being conducted on-line, which usually yields a low response rate [56]. Secondly, the cross-sectional nature of these data limits the extent to which causal inferences may be made. Thirdly, physical activity during lockdown was not assessed. However, it is the first study of this kind in the Caribbean region to understand the perceived stress, learning behaviour, and eating disorder among medical students during the COVID-19 pandemic. Another limitation of the current work is that we performed only bivariate analysis considering males and females separately. Considering the outcomes of learning behaviour and eating behaviour in a multivariate setting would reduce the biases and strengthen the quality of the study.

## 5. Conclusions

The effect of COVID-19 pandemic has transformed medical education and is likely to have long-lasting effects on student learning, mental well-being, and eating behaviour. The findings demonstrated that a substantial number of students had learning behaviour problems and a very high possibility of having eating disorder symptoms. The policymakers of the academic institutions should take appropriate measures and develop strategies to support learning and improve the mental well-being and eating behaviour of the students.

**Author Contributions:** Conception and design study, S.R.; Data collecting, S.R., A.P., N.N., S.A. and J.N.N.; Analysing and data interpretation, A.M.F.R. and S.R.; Original draft, S.R., R.K., B.R., M.T.V.M., A.P., N.N., S.A., J.N.N. and A.M.F.R.; Review and editing, S.R., R.K., B.R. and A.M.F.R. All authors have read and agreed to the published version of the manuscript.

**Funding:** This research received no external funding.

**Institutional Review Board Statement:** Ethical approval was obtained from the research committee of the American University of Integrative Sciences (AUIS-RC/2021/R#02-SR). The study was conducted by the guidelines of the 1975 Declaration of Helsinki.

**Informed Consent Statement:** Participants were fully informed about the nature of the study. The consent form explained the purpose of the study to the participants. Participation was voluntary, with participants given the option to withdraw at any time during data collection. The questionnaire was anonymous, with no identifying information being collected.

**Data Availability Statement:** Data and copies of the questionnaire are available upon reasonable request to the corresponding author.

**Acknowledgments:** The authors would like to thank the students who completed the questionnaire. This paper is dedicated to the memory of our colleague and co-researcher, Frank Kunik (Associate Professor of Human Physiology and Clinical Medicine, Dean of Student Affairs, AUIS), who suddenly passed away while the study was in progress. May his soul be in peace by the mercy of his Creator.

**Conflicts of Interest:** The authors declare no conflict of interest.

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
