# Peer review of "The Effects of Learning and Eating Behaviours among Medical Students during the COVID-19 Pandemic"

_ime, doi:10.3390/ime2020007_

Round 1
Reviewer 1 Report
A very important topic and I commend the authors for selecting the title of the research. However, I have the following comments:
· 1. Include a short streamline introduction on COVID-19 origin and transmission (refer and cite: doi: 10.1136/postgradmedj-2020-138234)
· 2. Since the study was conducted in an integrative set up, include a statement/paragraph on their daily routine food choices as per their country background. Did they usually consume fatty foods from childhood or was the choices available in the school/place of education was mainly fatty foods?
· 3. How many students declined to take part in the study? What were the main reasons?
· 4. Was the questionnaire validated and introduced to a pilot population before subjecting to wider participants?
· 5. The variables in your study should be divided into dependent and independent variables so the statistical variance can be estimated. This would identify the bias and reducing the bias would further enhance the rigidity of your study results.
· 6. Where other questionnaires apart from SCOFF looked upon? rephrasing some of its questions could improve the diagnostic validity of the SCOFF without compromising its brevity.
Reviewer 2 Report
This is an interesting topic which has been studied generally well. I think that in general the paper is in good shape. However, it needs to be better justified.
Having said that, the link between COVID and learning/ eating behaviours is lacking because COVID is presented as the only culprit here, when there are underlying psychological challenges due to studying medicine. There is a lot of evidence showing that medical students have the highest rates of mental illness among all other University students and people of same age from the general population. So COVID has basically added to the problem. I am not suggesting that the whole project should change. Instead, I suggest enriching the Introduction with information from the existing literature about the mental challenges and further discuss it the relation with the findings in the Discussion. Such knowledge/ information would better justify the need for this study and would give it broader scientific clout.
